# FIB-Assisted Fabrication of Single Tellurium Nanotube Based High Performance Photodetector

**DOI:** 10.3390/mi13010011

**Published:** 2021-12-22

**Authors:** Wangqiong Xu, Ying Lu, Weibin Lei, Fengrui Sui, Ruru Ma, Ruijuan Qi, Rong Huang

**Affiliations:** 1Key Laboratory of Polar Materials and Devices, Ministry of Education, Department of Electronic Science, School of Physics and Electronic Science, East China Normal University, Shanghai 200241, China; wangqiongx@126.com (W.X.); yinglu18@126.com (Y.L.); weibinleilove@126.com (W.L.); sfr19990206@126.com (F.S.); maru0828@163.com (R.M.); rhuang@ee.ecnu.edu.cn (R.H.); 2College of Physics and Electronic Engineering, Qujing Normal University, Qujing 655011, China

**Keywords:** FIB-assisted, Te nanotube, photodetector, optoelectronics

## Abstract

Nanoscale tellurium (Te) materials are promising for advanced optoelectronics owing to their outstanding photoelectrical properties. In this work, high-performance optoelectronic nanodevice based on a single tellurium nanotube (NT) was prepared by focused ion beam (FIB)-assisted technique. The individual Te NT photodetector demonstrates a high photoresponsivity of 1.65 × 10^4^ AW^−1^ and a high photoconductivity gain of 5.0 × 10^6^%, which shows great promise for further optoelectronic device applications.

## 1. Introduction

As one of the most important types of optoelectronic devices, photodetectors, (including photodiode, photoconductor and phototransistor), converting incident light into an electrical signal, have attracted great attention in the fields of flame sensing, ozone sensing, conversion communication, environmental monitoring, video imaging, night vision, material recognition, early detection of primary tumors and astronomical exploration because of their excellent photoelectrical properties [1,2,3,4]. Low dimensional Te nanomaterials, including Te nanowires (NWs) [5], Te nanobelt [6], Te nanotubes [7], and Te nanorods (NRs) [8] have been widely utilized as nonlinear optical response devices, optoelectronic devices, thermoelectric devices, piezoelectric devices, photonic crystal, self-developing holographic recording devices, radiative cooling devices, field-effect devices, and infrared acousto-optic deflector [9]. At present, the photoelectric properties of the low dimensional Te nanomaterials, including photoelectric conversion efficiency, photoconductivity gain and light responsivity, have attracted many researchers’ attentions, due to their high surface-area-to volume ratios and the reduced dimension of the effective conductive channel. Zhong et al. [10] prepared a photodetector based on the Te NWs with the photocurrent and responsivity reaching 0.17 mA and 25.8 AW^−1^, respectively. Kang et al. [6] reported that a Te nanobelt photodetector fabricated in-situ on SrTiO_3_ (001) substrate with the responsivity of 254.2 AW^−1^ under bias voltage of 1 V. Flexible photodetectors based on Te NRs were prepared by Qi Xiao et al. [11] with a responsivity of 6.1 AW^−1^. Zhang et al. [12] have reported the self-powered photoresponse behavior (2.79 µAW^−1^) of Te nanosheets irradiated by light at 400 nm under bias voltage of 0.6 V. Nevertheless, the photoelectric performance of the Te nanoscale devices has not been significantly improved. For example, their photocurrent and light responsivity are small, which are not suitable for practical applications. Moreover, the methods reported for the fabrication of Te nanodevices suffer from some disadvantages, such as low successful probability, nanomaterials wasting, complicated fabricating process or low yield, which hinder the applications of Te semiconductor nanomaterials in the photoelectric field. In the process of the single nanotube photodetector fabrication, it is important for the nanotube photodetector to realize reliable single nanotube electrical contact. There are some potential methods to do this work, such as dielectrophoresis [13] and Focused ion beam (FIB) etc. [14]. Compared with the other methods, FIB-assisted method is easy to implement to form ohmic contact and reliable electrical contact between Te nanomaterials and the electrodes with the Pt or W gas injected system (GIS), as both Pt or W possess very good conductivity.

In this work, Te nanotubes (NTs) with different diameters are synthesized by solvothermal method. Further, photodetector based on an individual Te nanotube was rapidly prepared by FIB-assisted technique, which is easy to implement with high successful probability and repeatability. The photodetector based on the individual nanotube with a diameter of 40 nm demonstrates a high photoresponsivity of 1.65 × 10^4^ AW^−1^ and a high photoconductivity gain of 5.0 × 10^6^%.

## 2. Experimental Section

### 2.1. Sample Preparation

Te nanotubes with different diameters are synthesized by solvothermal method. All the raw materials are commercially available and used without further purification. First, the tellurium powder (17.4 mg) and hydrazine hydrate (HH) (0.25 mL or 2.55 mL, 85%) were mixed in a 50 mL beaker, and then 40 mL deionized water was added into it. Next, the mixed solution was placed in the Teflon-lined autoclave (50 mL) and heated at 140 °C for 6 h in an oven. After cooling down to room temperature, the obtained products were cleaned by DI water and anhydrous alcohol for three times, respectively. Finally, the prepared powder was dried in a vacuum oven at 60 °C.

### 2.2. Sample Characterization

The crystal structures of as-synthesized samples were characterized by X-ray diffraction (D8tools, Bruker, Karlsruhe, Germany) with Cu Ka radiation. The microstructure was characterized by Raman spectra using a micro-Raman spectrometer (Jobin Yvon LabRAM HR 800UV, Paris, France) with a 532 nm laser source. Scanning electron microscopy (SEM) and energy dispersive spectrometer (EDS) analysis were performed on Zeiss Gemini 450 equipped with an X-ray energy dispersive spectrometer (Ultim Extreme, Oxford, UK). The microstructure was further investigated by high-resolution transmission electron microscopy (JEM-2100F, JEOL, Tokyo, Japan). For atomic resolution imaging, the measurements were performed on an aberration-corrected high-angle annular dark field scanning transmission electron microscopy (JEM Grand ARM300F, JEOL, Tokyo, Japan). To improve the signal-to-noise ratio (SNR) and to minimize the drift and the image distortion of high-angle annular dark field (HAADF) images, 10 serial frames were acquired with short dwell time (2 μs pixel^−1^). The image series were then aligned and superimposed. 

### 2.3. Fabrication of Individual Te Nanotube Photodetector Assisted by FIB Technique

As illustrated in Figure 1, the individual Te nanotube photodetector was fabricated by dual beam FIB (Helios G4, Thermo Fisher Scientific, Waltham, MA, USA)-assisted technique. First, Te nanotubes were dispersed in anhydrous alcohol and then transferred on a clean Si wafer by pipette. Then after the Si wafer was put into FIB system, single Te nanotube was extracted and transferred onto the 10 nm Cr/100 nm Au electrodes with a gap of 10 μm which were prepared by photolithography on SiO_2_/Si, by using nano-manipulator under SEM mode. In detail, the tungsten (W) probe of the nano-manipulator was inserted and moved to one end of the single Te nanotube. Then, Pt was deposited to contact W probe and Te nanotube under the acceleration voltage of 16 kV and the current of 0.23 nA to avoid the ion beam damage to Te nanotube. Next, the extracted Te nanotube was transferred on Au electrodes. Then, Pt pads were deposited at the contact between the Te nanotube and the Au electrodes under ion beam with the acceleration voltage of 30 kV and the current of 0.1 nA to eliminate the Schottky barriers between Te nanotube and Au electrodes and form high quality ohmic contacts [15].

## 3. Results and Discussion

### 3.1. Structural Analysis of Te Nanotubes

Figure 2a plots XRD patterns of the samples obtained in 0.25 mL and 2.55 mL HH. It can be seen that the as-prepared samples exhibit similar XRD patterns, and all prominent peaks can be indexed as the hexagonal tellurium with lattice constants of a = 0.450 nm and c = 0.592 nm, which matches well with the standard values of a = 0.446 nm and c = 0.593 nm (JCPDS Card No. 36-1452). As shown in Figure 2b, the Raman spectra of the samples features three dominant active phonon modes due to their large electronic polarizability. For the sample obtained in 0.25 mL HH, the weak peak at 90.8 cm^−1^ is related to the E^1^ mode of Te which is caused by the rigid-chain rotation over the a- and b-axis [16]. The peak at 119.8 cm^−1^ is related to the A^1^ mode of Te, corresponding to the chain expansion mode in which each atom moves in the basal plane. The peak at 139.2 cm^−1^ related to the E^2^ mode of Te is mainly related to the asymmetric stretching along the c-axis [17,18]. Moreover, there is a second-order weak peak at 268.9 cm^−1^ which is related to the second order harmonic of the E mode of Te [19]. The E^1^, A^1^ and E^2^ modes are at the peak of 92.8, 120.8 and 141.1 cm^−1^, respectively, for the sample obtained in 2.55 mL HH, which has a small blue shift compared with the sample obtained in 0.25 mL HH. The observed blue shift is determined by the diminish of the intrachain bonds (covalent bonds) and the increase of the interchain interactions (van der Waals) [20]. It means that the sample synthesized in 2.55 mL HH might have stronger intrachain interactions (van der Waals) than that in 0.25 mL HH.

### 3.2. Morphology and Microstructure of Te Nanotubes

As shown in Figure 3a–d, it can be seen that the samples prepared with 0.25 mL and 2.55 mL HH are Te nano/micro tubes with different diameters (nanotube: around 70 nm in 0.25 mL HH, microtube: around 3 μm in 2.55 mL HH). Representative HRTEM images of the Te nano/micro tubes with diameter of around 70 nm and 4 μm are shown in Figure 3b,d, illustrating obvious bright and dark lattice fringes with the lattice spacings of 0.59 nm corresponding well with the lattice spacings of (001) planes for hexagonal tellurium, which matches well with the XRD results. The above HRTEM results confirm that the lattice fringe (001) is along the longitudinal axis of the Te tubes, which suggests that the Te N/MTs exhibits a preferential growth direction along the [001] zone axis (c-axis). Further, Figure 3f is typical atomic STEM-HAADF image of the Te nanotubes projected along the [001] zone axis, which fit well with the atomic simulated models (Figure 3e) using the P3121 space group.

### 3.3. Photoelectric Properties of Individual Te Nanotube Photodevice

The photoelectric properties of Te nanotubes are critical to their applications in the field of the photoconductive devices. In this work, photodevice based on a single Te NT with a diameter of around 70 nm was prepared by FIB-assisted method (the detailed procedure refers to the Experimental Section) with the SEM image of the Te-nanotube based photodetector shown in the inset of Figure 4a. The photoresponse of an individual Te nanotube photodevice was explored by studying on their output characteristics of photocurrent at room temperature using a two-probe method by a Keithley 4200 system with 405 nm UV light [9]. Figure 4b displays the I-V characteristics of the photodetector based on the Te NT under the irradiation of a wavelength of 405 nm with different power densities. With power density varying from 0.6 to 1.2 W/cm^2^, the photocurrent monotonically increases, demonstrating an enhanced photoconductivity. Furthermore, the I-V curves bent obviously at the voltage of 0.3 V, suggesting the formation of the schottky contact between the Te NT and the Cr/Au electrodes. The photocurrent and dark current of the photodetector based on the Te NT reach 1.42 mA and 4.29 μA, respectively, at a voltage of 0.6 V under the power density of 1.2 W/cm^2^. Figure 4c shows that the photocurrent is symmetric.

Figure 4d shows that the photocurrent responses increase steadily with increasing irradiation density. High power densities provide the Te NT with more photons, thus generating more non-equilibrium carriers in Te, and subsequently resulting high photocurrents. The dependence of the photocurrent on irradiance density can be fitted by the power law [21]: I_P_ = aP^θ^, where θ is a parameter related to the trapping and recombination processes of the photocarriers in the photodetectors. Obviously, for Te NT, the photocurrent increases with an increasing power density, which is consistent with the results that the charge carrier photogeneration efficiency is proportional to the absorbed photon flux [22]. The θ obtained for the Te NT photodetector is 0.477 which is related to the processes of electron-hole generation, trapping and recombination [23,24].

The spectral responsivity (R), a vital parameter of a photodetector, is defined as the photocurrent generated by the irradiation on the effective area of Te and can be expressed as [25,26,27]: R = I_p_/PA, where I_P_ = I_photo_ − I_dark_ ≈ I_photo_, I_dark_ is the dark current, P is the incident light intensity irradiated on the single Te, and A is the effective illuminated area of the single Te. The photoconductive gain (G), another important performance parameter, is mainly defined as the number of carriers circulating through a photodetector per absorbed photon and per unit time and can be expressed as [28,29]: G = (I_p_/e)/(PA/hʋ), where I_p_ is the photocurrent, P is the incident power density, A is the effective irradiated area on the Te, hʋ is the energy of an incident photon, and e is the electronic charge. Figure 4e shows the R and G values of a single Te NT photodetector reach 1.65 × 10^4^ AW^−1^ and 5.0 × 10^6^%, respectively, at a voltage of 0.6 V and under the irradiation of a wavelength of 405 nm. The curves of the spectral responsivity (R) and the photoconductivity gain (G) show a downward trend with the light power density increasing, as a high light power density could reduce the trapped states with an increasing photogenerated holes, resulting in the photocurrent saturation. The process can be described as follow: under low light power density, excited holes are trapped by the surface trap states and the crystal structure defects of the Te NT with the reduction of the photo-generated carrier recombination at the recombination center and the prolong of the lifetime of photo-generated carriers. When the Te NT is irradiated with high light power density, the increasing photo-generated holes fill the surface trap states and the crystal structure defects of the Te NT. The photogenerated carriers are easy to recombine at the recombination center and not take part in the charge transfer process [30] resulting in the small R and G values under the high light power density.

As summarized in Table 1, the photoelectric performance of the nanodetector based on a single Te NT in this work are enhanced a lot comparing with that of other 1D and 2D nanomaterials, which exhibits high I_photo_/I_dark_, photoconductivity gain (G), and spectral responsivity (R), demonstrating that this nanodevice shows an extremely high sensitivity to violent light and possesses high effective availability. The superior performance of the individual Te nanotube photodetector might attribute to large surface-to-volume ratio of the nanotube, good crystallinity.

## 4. Conclusions

In summary, Te nanotubes with different diameters with hexagonal phase growing along [001] direction were synthesized by solvothermal method. With the assistance of FIB technique, Te photodetector based on a single nanotube with diameter of 40 nm was successfully prepared, which exhibits very high responsivity (1.65 × 10^4^ AW^−1^) and photoconductivity gain (5.0 × 10^6^%). Our work provides a promising and effective way for the preparation of photodetectors based on one-dimensional Te nanostructures, which may further promote the application of one-dimensional semi-conductive nanostructures in the fabrication of high-performance optoelectronic nanodevices.

## Figures and Tables

**Figure 1 micromachines-13-00011-f001:**
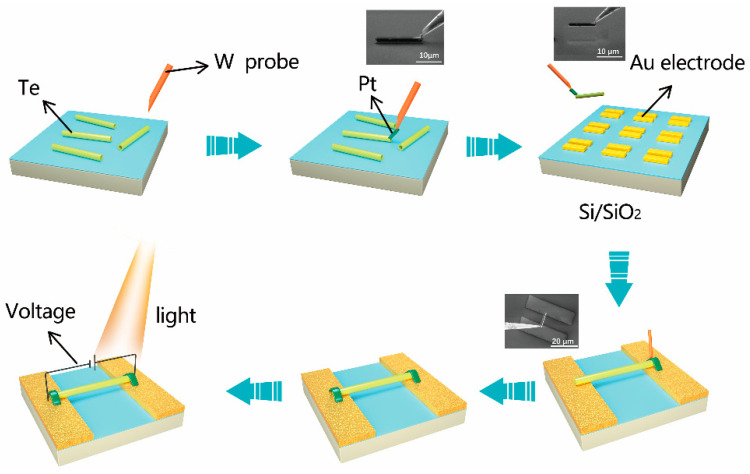
The schematic illustration of the fabrication process of individual Te nanotube photodetector assisted by FIB technique. Insets are SEM images taken during the fabrication process.

**Figure 2 micromachines-13-00011-f002:**
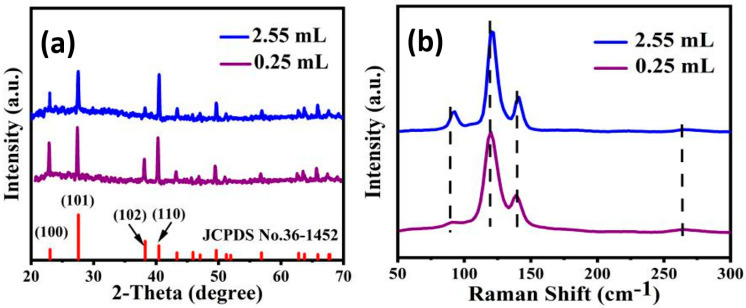
(**a**) XRD patterns and (**b**) Raman spectra of the as-prepared samples.

**Figure 3 micromachines-13-00011-f003:**
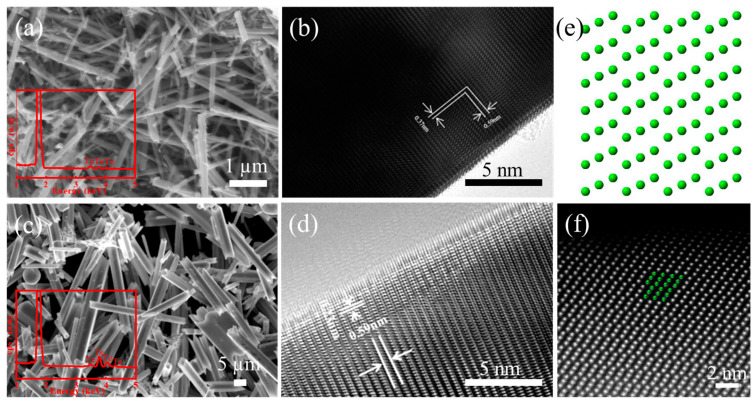
(**a**) SEM and (**b**) HRTEM images of the Te NTs obtained in 0.25 mL HH. (**c**) SEM and (**d**) HRTEM images of the Te NTs obtained in 2.55 mL HH. (**e**) Schematic crystal structure image of Te projected along [001] axis. (**f**) HAADF images of the Te NTs obtained in 2.55 mL HH projected along [001] axis.

**Figure 4 micromachines-13-00011-f004:**
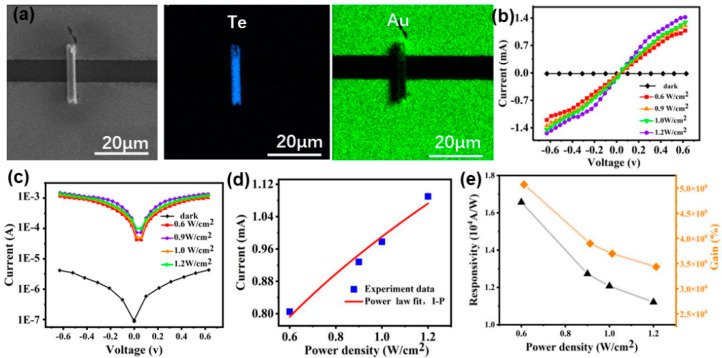
The photoresponse properties of the single Te NT detector: (**a**) SEM image and EDS maps of Te and Au; (**b**) I-V curve, (**c**) logarithmic plot of (**b**); (**d**) the relationship between the photocurrent and the optical power density; (**e**) photoresponsivity and photoconductivity gain.

**Table 1 micromachines-13-00011-t001:** Comparison of the spectral responsivity, the photoconductivity gain, I_photo_/I_dark_ and non-unity exponent θ of the Te NT with previous reported work.

Photodetector	I_photo_/I_dark_	θ (I_P_ = aP^θ^)	Gain (%)	Responsivity (AW^−1^)	References
Te NT	331	0.76	5.0 × 10^6^	1.65 × 10^4^	this work
Er-CdSe NB	2.21 × 10^4^	0.603	3.87 × 10^5^	2.17 × 10^3^	[31]
CdS FL	10^3^	0.76	55.87	0.18	[25]
Se NR	—	0.57	88.4	0.408	[32]
CdTe NS	27	—	—	6.0 ×10^−4^	[33]
PbS	<2	—	—	1.6 × 10^3^	[34]

Note: NB nanobelt, NS nanosheet, FL flake, NW nanowire, NR nanorod, I_dark_: dark current, I_photo_: photocurrent.

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
