# Peer review of "FIB-Assisted Fabrication of Single Tellurium Nanotube Based High Performance Photodetector"

_micromachines, 2021, doi:10.3390/mi13010011_

Round 1
Reviewer 1 Report
The manuscript is interesting and the results of experimental investigations are clear, but several points in discussion should be clarified:
Line 157-158, the formation of Schottky-barrier is stated from I-V curve. How does this barrier affect the photocurrent? In line 205 it is mentioned as a reason of high photoconductive gain.
Line 164, generation of many excitons results in high photocurrent. How can excitons contribute the current, if they are neutral?
Line 171, A high intensity could reduce the trapped states caused by an increasing photogenerated holes... This sentence is unclear. How can holes be increasing? The same question to line 190. And how can intensity reduce states? State reduction is usually charge exchange process, does intensity have charge?
As can be seen in line 180, the photoconductive gain (G) is roughly the number of electron hole pairs, excited by single photon. However, the value of G corresponds to 10^4 electron-hole pairs per photon. How can one photon with energy of 3 eV excite so many electron-hole pairs in Te, which has 0.35 eV band gap? It should be explained or the definition of the photoconductive gain should be clarified to match energy conservation law.
In line 204 high photocurrent response is attributed to high surface-to-volume ratio. This statement seems to be counterintuitive, because surface recombination should decrease the photocurrent, so, please clarify.
Author Response
Comment 1: Line 157-158, the formation of Schottky-barrier is stated from I-V curve. How does this barrier affect the photocurrent? In line 205 it is mentioned as a reason of high photoconductive gain.
Response: As reported, the work functions of Te, Cr, Au and Pt are around 5.64eV, 4.6eV, 5.1eV and 5.65eV. Because of the intrinsic p-type doping in crystalline Te, the difference in the work function between Te and Cr/Au electrodes will result in Schottky-barrier. Hence, only an ultralow current can pass through Te photodetector with a lower bias or under dark condition. Same phenomenon was observed in the recently report paper (Shen et al., Science 374, 1390–1394 (2021) )
In “Elemental electrical switch enabling phase segregation–free operation” (Shen et al., Science 374, 1390–1394 (2021) ), Zhu et al reported: Single-element tellurium (Te) volatile switch with a large (≥11 megaamperes per square centimeter) drive current density, ~103 ON/OFF current ratio, and faster than 20 nanosecond switching speed. The low OFF current arises from the existence of a ~0.95–electron volt Schottky barrier at the Te–electrode interface, whereas a transient, voltage pulse–induced crystal-liquid melting transition of the pure Te leads to a high ON current.
Revisions: In our work, we think the Schottky barrier will be beneficial to a low leakage current. We have changed the sentence in line 205 as: The superior performance of the individual Te nanotube photodetector might attribute to large surface-to-volume ratio of the nanotube, good crystallinity.
Comment 2: Line 164, generation of many excitons results in high photocurrent. How can excitons contribute the current, if they are neutral?
Revisions: Sorry for the mistake. We have changed this sentence as: High power densities provide the Te NT with more photons, thus generating more non-equilibrium carriers in Te, and subsequently resulting high photocurrents.
Comment 3: Line 171, A high intensity could reduce the trapped states caused by an increasing photogenerated holes... This sentence is unclear. How can holes be increasing? The same question to line 190. And how can intensity reduce states? State reduction is usually charge exchange process, does intensity have charge?
Response: Thank you very much for your kindly reminder. Sorry for this confusion.
Under lower power irradiation, the surface trap states and structure defects of Te nanotube will be occupied by photogenerated holes, and these holes will also be combined with the surface negative charge oxygen. When a higher power light source is used, the surface trap states and structure defects of Te nanotube will be completely filled owing to the increase of photogenerated hole populations. The newly generated holes will in turn combine with electrons promptly, and thus both electrons and holes will not participate in charge transfer process.
Revisions: We have changed our description in this manuscript for better understanding: The curves of the spectral responsivity (R) and the photoconductivity gain (G) show a downward trend with the light power density increasing, as a high light power density could reduce the trapped states with an increasing photogenerated holes [25], resulting in the photocurrent saturation. The process can be described as follow: under low light power density, excited holes are trapped by the surface trap states and the crystal structure defects of the Te NT with the reduction of the photo-generated carrier recombination at the recombination center and the prolong of the lifetime of photo-generated carriers. When the Te NT is irradiated with high light power density, the increasing photo-generated holes fill the surface trap states and the crystal struc-ture defects of the Te NT. The photogenerated carriers are easy to recombine at the re-combination center and not take part in the charge transfer process [30] resulting in the small R and G values under the high light power density.
Comment 4: As can be seen in line 180, the photoconductive gain (G) is roughly the number of electron hole pairs, excited by single photon. However, the value of G corresponds to 10^4 electron-hole pairs per photon. How can one photon with energy of 3 eV excite so many electron-hole pairs in Te, which has 0.35 eV band gap? It should be explained or the definition of the photoconductive gain should be clarified to match energy conservation law.
Response: Thank you very much for the kindly reminder.
As commonly reported, photoconductive gain, meaning the quantum efficiency (QE) (defined as the number of carriers circulating through a photodetector per absorbed photon and per unit time) determines the efficiency of an electron transport and carrier collection. Photodetector based on individual In2Ge2O7 nanobelts showed high responsivity (3.9 ×10 5A W − 1) and quantum efficiency (2.0×108%), which can be ascribed mainly to surface traps, one-dimensionality, and high-quality single-crystal character (Adv. Mater. 2010, 22, 5145–5149). GaN nanowire (band gap of 0.342eV) optoelectronic nanodevice exhibits an outstanding performance in monitoring the irradiation of UV-A rays with high photosensitivity (2 × 104%), ultrahigh photoresponsivity of 1.74 × 107 A/W, and EQE of 6.08 × 109 % (ACS Appl. Mater. Interfaces 2017, 9, 2669−2677).
Revisions: Related changes have been made in the manuscript: “The photoconductive gain (G), another important performance parameter, is mainly defined as the number of carriers circulating through a photodetector per absorbed photon and per unit time”
Comment 5: In line 204 high photocurrent response is attributed to high surface-to-volume ratio. This statement seems to be counterintuitive, because surface recombination should decrease the photocurrent, so, please clarify.
Response: Thank you very much for your advice.
Compared with their bulk peers, one-dimensional (1D) semiconductor nanostructures exhibits smaller size, huge surface-to-volume ratio, decent crystal quality and high light extraction/ absorption efficiency, which means the defect density of 1D semiconductor nanostructures is much smaller than that of the bulk materials. So, we think high surface to volume ratio will not results in high surface defect state.
It is acceptable that high surface-to-volume ratio will be beneficial to enhance the quantum efficiency and shorten the response time of photodetectors.

Reviewer 2 Report
In their work, Xu and co-authors detail the fabrication and evaluation of Te nanotubes through FIB method. They find a high photoresponse and photoconductivity gains, and suggest this as a method to prepare future devices. I believe their technical results to be sound, and have only a few minor issues I would like the authors to address before publication.
- In Fig. 4, there must be an error, correct? The scale bar indicates the rod diameter is on the order of 3-4 nm and the electrode gap is on the order of 10 nm. The rods seem much larger in Fig. 3. Further, placement of a rod accurately on a gap this small, using the FIB technique seems difficult at best. I believe this is a scale bar issue, and it should be in um, not nm? The same issue is present in Fig. 1 (schematic of the process.
- 405 nm light was used for the photoresponse measurements. Can the authors clarify why this specific wavelength was used, and if there should be a significant wavelength dependence?
- The fabrication process for the single nanorod measurements required a Pt deposition step at relatively high energies. Does Pt implantation into the Te rods play a significant role, or can this be neglected?
Author Response
Comment 1: In Fig. 4, there must be an error, correct? The scale bar indicates the rod diameter is on the order of 3-4 nm and the electrode gap is on the order of 10 nm. The rods seem much larger in Fig. 3. Further, placement of a rod accurately on a gap this small, using the FIB technique seems difficult at best. I believe this is a scale bar issue, and it should be in um, not nm? The same issue is present in Fig. 1 (schematic of the process.
Response: Thank you very much for the suggestions. We have changed the error. Right scale bars were used in the new images.
Comment 2: 405 nm light was used for the photoresponse measurements. Can the authors clarify why this specific wavelength was used, and if there should be a significant wavelength dependence?
Response: As reported, Te-based photodetectors show broad-spectral response in a wide wavelength from near-ultraviolet to near-infrared regime. In our work, limited by the experimental conditions, commonly used wavelength of 405nm was used to estimated the properties of the photodetector.
Comment 3: The fabrication process for the single nanorod measurements required a Pt deposition step at relatively high energies. Does Pt implantation into the Te rods play a significant role, or can this be neglected?
Response: In FIB system, GISs are commonly used to deposition of C/Pt/W protection layers under E beam or I beam. The deposition process is like an E beam or I Beam assisted CVD process, where very small amount of gas (organometallics: Pt(CO)6 or W(CO)6) is ejected from the needle of GIS, and then the organometallics will decompose to Pt or W under E beam or I beam. The energy needed for the decomposition of the organometallics is very low. The deposition of Pt or W only occurred at the top surface of the samples, which may not lead to implantation effect.
